# Management of Acute Life-Threatening Asthma Exacerbations in the Intensive Care Unit

**Thomas Talbot [1], Thomas Roe [1] and Ahilanandan Dushianthan [1,2,3,]**

1  General Intensive Care Unit, University Hospital Southampton, Southampton SO16 6YD, UK;
   thomas.talbot@uhs.nhs.uk (T.T.); thomas.roe@uhs.nhs.uk (T.R.)
2  National Institute for Health and Care Research (NIHR) Biomedical Research Centre, University Hospital
   Southampton, Southampton SO16 6YD, UK
3  Clinical and Experimental Sciences, Faculty of Medicine, University of Southampton,
   Southampton SO16 6YD, UK
*  Correspondence: a.dushianthan@soton.ac.uk

**Abstract:** Managing acute asthma exacerbations in critical care can be challenging and may lead to adverse outcomes. While standard management of an acute asthma exacerbation is well established in outpatient and emergency department settings, the management pathway for patients with life-threatening and near-fatal asthma still needs to be fully defined. The use of specific interventions such as intravenous ketamine, intravenous salbutamol, and intravenous methylxanthines, which are often used in combination to improve bronchodilation, remains a contentious issue. Additionally, although it is common in the intensive care unit setting, the use of non-invasive ventilation to avoid invasive mechanical ventilation needs further exploration. In this review, we aim to provide a comprehensive overview of the available treatments and the evidence for their use in intensive care. We highlight the ongoing need for multicentre trials to address clinical knowledge gaps and the development of intensive-care-based guidelines to provide an evidence-based approach to patient management.

**Keywords:** asthma; intensive care; ventilation; non-invasive ventilation

## 1. Introduction

Asthma is one of the most common respiratory conditions encountered by physicians across all specialties. According to global estimates, asthma affects around 260 million people and caused 455,000 deaths in 2019 worldwide [1]. In the United States (USA), 13% of the population suffers from lifetime asthma with significant associated socio-economic burden [2]. Asthma is commonly managed in the primary care setting or during acute exacerbations in the emergency department (ED). However, severe exacerbations require hospitalisation, and the most severe or life-threatening exacerbations may require admission to the intensive care unit (ICU) and the initiation of mechanical ventilation [3]. According to the latest statistics from Public Health England for those ages 19 years or older, there were 21,125 admissions to EDs across England in the financial year 2020/21, with a median hospital stay of 2 days and an overall asthma mortality rate of 2.36 per 100,000 people. The childhood asthma data showed that there were 16,310 admissions to EDs, with a median length of stay of 1 day [4].

Despite the rising prevalence, there seems to be an improvement in the hospital admissions associated with life-threatening asthma. The ICU admission rate for hospitalised asthma patients ranges from 1–10% [5,6]. In a comprehensive analysis of almost two million hospitalisations for asthma in the United States, the healthcare utilisation patterns were examined and compared between 2000 and 2008. The findings revealed that the proportion of cases requiring invasive mechanical ventilation (IMV) decreased from 1.40% in 2000 to 0.73% in 2008 [7]. However, there was a fivefold increase in the utilisation of non-invasive ventilation [7]. Among the patients with asthma admitted to the ICU, the incidence of IMV

requirement is estimated to be between 15% and 56% depending on the case series [5,6,8]. According to the Intensive Care National Audit and Research Centre (ICNARC), the United Kingdom (UK) ICU case mix programme, which analysed over 800,000 ICU admissions in the UK from 2002 to 2011, found that acute asthma constituted approximately 1.40% of all ICU admissions. Moreover, the proportion of patients requiring IMV within 24 h of admission was 46%, with an overall hospital survival rate of 93% [9].

Acute severe asthma exacerbations leading to ICU admission require immediate assessment and the institution of management strategies to minimise the morbidity and burden associated with mechanical ventilation. Most acute asthma management guidelines do not go far enough to detail specific ICU interventions [10–12], and consequently, in this review, we aim to evaluate the assessment and management of patients with acute asthma exacerbation admitted to the ICU. This review will focus primarily on management strategies for severe, life-threatening, and near-fatal asthma exacerbations in adults.

## 2. Classifications of Asthma Severity

It is worth noting that the definition of what constitutes critical care can differ widely around the world, with an interesting dichotomy between the UK and other countries such as the USA and those in Western Europe. Critical care facilities per capita are fewer in the UK than in the USA and Western Europe and generally cater to a patient population who are more unwell and have a higher mortality. Figure 1 is a summary of available potential treatments and clinical features of acute asthma with varying severity according to the British Thoracic Society (BTS)/Scottish Intercollegiate Guidelines Network (SIGN) and potential treatment strategies available in the management of a patient in the ICU [10]. Management in the ED is comprehensively covered within the BTS/SIGN guidelines already and will not be discussed at length [10].

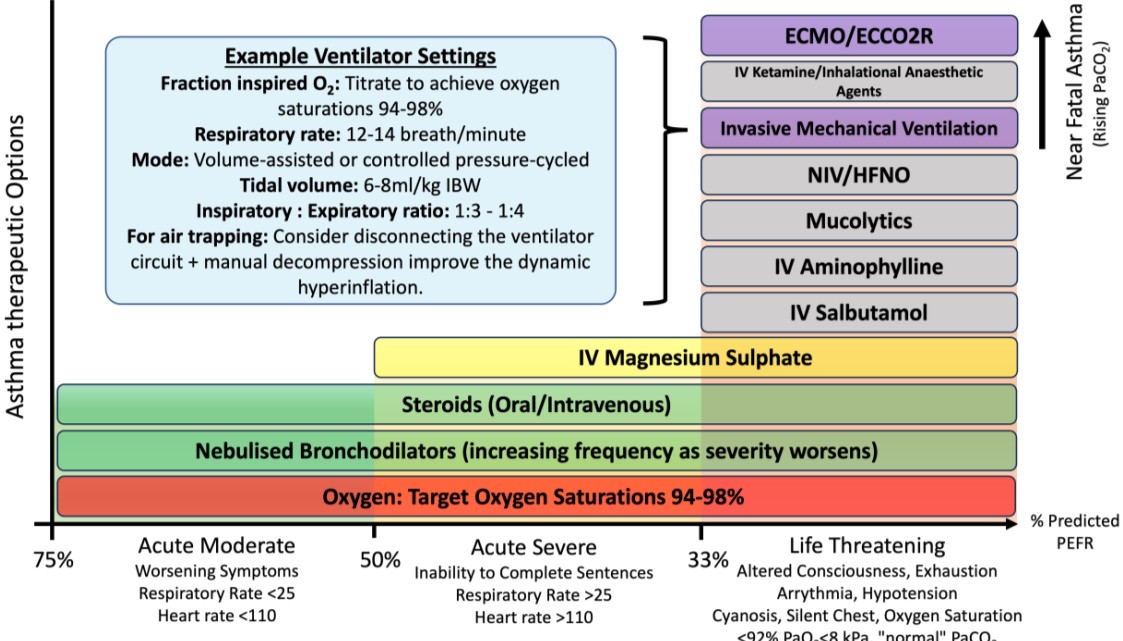

**Figure 1.** Therapeutic options for acute moderate-near fatal asthma [10]. The *X*-axis represent the severity of asthma exacerbation, and the *Y*-axis portrays the potential therapeutic options available. *Y*-axis does not represent order of treatment in life-threatening column as this is not clearly defined in current guidelines. Coded as per the quality of the evidence supporting clinical use (green: high, amber: moderate, red: low, grey: insufficient evidence for recommendation, purple: unclear but considered standard practice as rescue therapies given that randomised trials are not ethically feasible). IV: intravenous. NIV: non-invasive ventilation. HFNO: high-flow nasal oxygen. ECMO: extracorporeal membrane oxygenation. ECCO2R: Extracorporeal membrane caron dioxide removal.

## 3. Management of Acute Asthma Exacerbation

Acute severe asthma exacerbations can very rapidly evolve into near fatal asthma, and it is imperative that the critical care team makes a swift and thorough assessment of all referred patients. A systematic approach to assessing the wide range of physiological derangements for asthma with institution of appropriate management is fundamental to avoiding progression to near fatal asthma and subsequent need for mechanical ventilation.

### 3.1. Oxygen Therapy and Targets

Although asthma is primarily a large airways disease, some patients with acute asthma exacerbation may present with hypoxemia of varying severity, and targeted oxygen therapy is recommended. The BTS guidelines currently recommend an oxygen saturation of 94–98% [13]. However, there is contention in the literature surrounding the level at which oxygen targets should be set for critically ill patients, with concern regarding the effects of hyperoxemia on deleterious outcomes [14]. A randomised controlled trial (RCT) of 106 patients presenting to the ED with acute severe asthma concluded that high-concentration oxygen therapy (8 L/min via a medium-concentration Hudson facemask) significantly increases $PaCO_2$ and recommended a titrated regime where oxygen is only administered to patients suffering from hypoxemia with an oxygen saturation of ≤92% [15].

The 2018 Improving Oxygen Therapy in Acute-illness (IOTA) systematic review and meta-analysis reviewed 16,037 hospitalised patients with various acute conditions and highlighted that hyperoxemia may be associated with adverse outcomes [16]. In this study, liberal oxygen therapy was associated with an increased in-hospital mortality. Several other studies have provided conflicting evidence [17,18] regarding ideal oxygen targets, and as such, guidelines for its use in acute illnesses, including asthma, still need to be updated [19,20]. To draw definitive conclusions, additional high-quality RCTs are required to understand the physiological effects of oxygen, especially during acute asthma exacerbations. The UK-ROX and MEGA-ROX trials are multi-centre trials aiming to recruit a large number of ICU patients (16,500 and 40,000, respectively) to identify the ideal oxygen saturation target for critically ill mechanically ventilated patients [21,22]. These trials should provide important insight into an intervention previously regarded as a cornerstone of management for a wide range of conditions. Sub-group analysis of these large trials may give additional guidance specific to asthmatic patients in the ICU. While waiting for further evidence, it is essential to avoid both hypoxemia and hyperoxemia, and supplementing inspired oxygen to the current recommended oxygen saturation target of 94–98% appears safe and appropriate [10,13].

### 3.2. Nebulised Bronchodilators

Nebulised β2 adrenoceptor agonists (salbutamol/albuterol) are central to the treatment of acute asthma exacerbations of all severity as well as for day-to-day maintenance. β2 adrenoceptors are G-protein-coupled transmembrane receptors that exist predominantly in airway smooth muscle cells. They activate the enzyme adenylyl cyclase, which produces cyclic adenosine monophosphate (cAMP) that likely activates protein kinase A and modification of intracellular calcium concentrations. Differences in the activity of β2 agonists mediate the kinetics of how airway smooth muscle responds; for example, salbutamol directly activates the adrenoceptor, while salmeterol interacts with a receptor-specific auxiliary binding site [23]. The first-line therapy for rapid bronchodilation requires the use of high-dose nebulised $β_2$ agonists driven by either oxygen or air depending on the degree of hypoxemia. Patients with minimal response to the initial dose may require continuous or "back-to-back" nebulisation. Intravenous β2 agonists are reserved for those patients in whom the nebulised therapy cannot be used reliably or in those who require additional bronchodilation during extreme states and are discussed in detail in the later section. The adverse effects of β2 adrenoceptor agonists are well known and pharmacologically predictable. They primarily include tachycardia and tremor but also have effects on serum

potassium and glucose. These effects tend towards tolerance with continued exposure, although this is of little relevance in the acute exacerbation phase [24].

### 3.3. Systemic Corticosteroids

The beneficial effects of corticosteroids in the management of acute asthma have long been known [25]. A 2001 Cochrane review including 12 studies and 863 patients concluded that corticosteroids given within the first hour of presentation to the ED significantly reduced the rate of hospital admission in patients presenting with acute asthma [26]. Corticosteroids reduce mortality, asthma relapses, subsequent hospitalisation, and requirement for β2 agonist therapy. Since they take time to work, the earlier they are given in the acute attack, the better the outcome [27]. The BTS guidelines advise giving steroids in adequate doses to all patients with acute asthma exacerbation [10]. Regarding the route by which steroids are given, studies have shown that the efficacy between oral, intravenous, and intramuscular routes is similar. Furthermore, there appears to be little difference in efficacy between different formulations of corticosteroids—for example, prednisolone, dexamethasone, and methylprednisolone [28–31]. A standard dose of 40–50 mg of prednisolone or 400 mg of hydrocortisone (100 mg four times a day) for a minimum duration of 5 days is currently recommended [10]. Although there are several acute side effects from systemic corticosteroids use in ICUs, including hyperglycaemia, hypernatremia, water retention, delirium, and ICU-acquired weakness [32], the benefits of short-term modest doses of systemic corticosteroids outweigh these adverse events.

### 3.4. Magnesium Sulphate

Intravenous (IV) and nebulised magnesium sulphate ($MgSO_4$) have long been proposed as a treatment for severe acute asthma. IV $MgSO_4$ is commonly used in the initial stages of refractory severe asthma in the ED, but there is little literature on its use or effectiveness beyond this stage. The mechanism of action of $MgSO_4$ is only partially understood but is thought to comprise the following: inhibition of the cellular uptake of calcium via activation of $Na^+Ca^{2+}$ pumps, muscle relaxation via inhibition of myosin and calcium interaction and calcium-dependent acetylcholine release at motor neuron terminals, and inhibition of prostacyclin and nitric oxide synthesis. Furthermore, there appears to be a degree of anti-inflammatory response via inhibition of mast cell degranulation, T-cell stabilisation, and the attenuation of an individual's neutrophil respiratory burst rate, provided by the inhibition of calcium uptake. The overall result of these effects is to increase bronchodilation and reduce the severity of acute asthma [33–36].

The 3 Mg Trial randomised 1109 adult patients presenting to EDs with acute severe asthma (as defined by BTS guidelines), with those suffering life-threatening features excluded [37]. Participants were randomised to IV $MgSO_4$ therapy, nebulised $MgSO_4$ therapy, or standard therapy alone. The study did not demonstrate any clinical benefits from either form of magnesium therapy. Even though there was evidence of some beneficial effect on subsequent rehospitalisation, there did not appear to be any effect on dyspnoea (via the visual analogue scale) or on peak expiratory flow rate (PEFR) when compared to a placebo. There was no evidence of improvement from a placebo with nebulised $MgSO_4$ [37]. A Cochrane review and meta-analysis of 13 studies including 2313 patients concluded that "a single infusion of 1.2 g or 2 g IV $MgSO_4$ over 15 to 30 min reduces hospital admissions and improves lung function in adults with acute asthma who have not responded sufficiently to oxygen, nebulised short-acting β2 agonists and IV corticosteroids" [38]. However, the effect of $MgSO_4$ on patients with life-threatening or near-fatal asthma in preventing the need for mechanical ventilation and mortality of mechanically ventilated ICU patients is not known. Currently, magnesium sulphate is recommended for those who have refractory symptoms despite standard initial management [10].

### 3.5. Intravenous Aminophylline

Methylxanthines can cause bronchial smooth muscle relaxation through multiple mechanisms and are one of the most commonly prescribed drugs for asthma worldwide [39]. The effect of methylxanthines is due to a combination of inhibition of phosphodiesterase enzymes, translocation of calcium, adenosine receptor blockade, and modulation of GABA receptors, resulting in the accumulation of cyclic adenosine monophosphate and subsequent bronchodilation [39–42]. There is a reasonably large body of literature supporting the use of oral theophylline in difficult asthma, but evidence for the use of intravenous aminophylline (a more soluble form of methylxanthine) as an adjunct in acute asthma exacerbation has remained controversial, and indeed its use has declined in recent decades [43]. However, intravenous aminophylline has a narrow therapeutic index and should be used with caution, as it requires monitoring of plasma concentrations to avoid toxicity.

While the guidelines currently do not recommend the routine use of intravenous aminophylline, it continues to be used in patients with severe asthma as an adjunct to provide additional bronchodilation [44,45]. RCTs in adults investigating the use of IV aminophylline in life-threatening or near-fatal asthma are sparse, and most studies were performed on small sample sizes. Currently, BTS recommends the use of oral theophylline for use in asthma only under specialist guidance [10]. Conversely, the guidelines state that intravenous aminophylline is unlikely to result in any additional bronchodilation compared with standard care with inhaled bronchodilators and steroids, and the guidelines remind clinicians of the increase in side effects such as vomiting and arrhythmias [10]. A meta-analysis involving seven trials and 380 children concluded that although there appeared to be an improvement in lung function as measured by PEF and $FEV_1$, they found that there was no apparent reduction in symptoms, number of nebulised treatments, or length of hospital stay [46]. A meta-analysis in 2012 of 11 studies involving 350 patients found no consistent evidence favouring either intravenous β2 agonists or intravenous aminophylline in the treatment of acute asthma. It went further to state that the heterogeneity of the trials included, alongside the increased risk of adverse events, should be considered given the lack of evidence surrounding the efficacy of these agents [47]. Although intravenous aminophylline is sometimes used as a salvage therapy, clinical trials or a large data registry are required to assess its effectiveness as an adjunctive treatment for patients with severe asthma in the ICU.

### 3.6. Intravenous Salbutamol

There are studies going back to the mid-1970s exploring the effect of IV β2 agonists for the treatment of acute asthma. However, it is unclear if IV β2 agonists offer additional bronchodilation beyond the use of regular nebulisation that leads to better outcomes. A meta-analysis conducted in 2001 involving 15 RCTs and 584 patients concluded that there was no significant evidence to support the use of IV β2 agonists over the nebulised form to improve bronchodilator response [48]. However only 20% (3/15) of the studies directly assessed the outcomes of using IV β2 agonists in addition to the nebulised form. It is also worth noting that these trials evaluated different age groups (adult vs. paediatric) and used different primary outcomes, and as such, only limited conclusions can be drawn from the pooled data. Hence, the role of adding the IV form to the nebulised treatment remains somewhat unclear.

A subsequent meta-analysis was published in 2012 specifically exploring the addition of IV β2 agonists to inhaled β2 agonists in acute asthma [47]. This analysis included three RCTs involving a total of 104 patients: Browne et al. (1997) [49] (29 children), Nowak et al. (2010) [50] (29 adults) and Bogie et al. (2007) [51] (46 children). Of these, only Browne et al. (1997) supported the use of the IV form with evidence of shorter recovery time and slight benefit regarding pulmonary index scores [49]. It was noted however, that these advantages should be considered carefully with respect to increased side effects associated with IV β2 agonists. Currently, there is no evidence to support the use of IV β2 agonists in patients with severe asthma in the ICU. However, it is available as an additional bronchodilator

treatment in those with severe airflow obstruction, and it continues to be used as a rescue therapy in ICU patients [45].

## 4. Acute Life-Threatening Asthma Advanced Management in the ICU

Patients with acute life-threatening and near-fatal asthma should be admitted to the ICU for close monitoring and to initiate additional measures to optimise bronchodilation and ventilation. However, subsequent management of such patients is generally not addressed by current guidelines due to a lack of high-quality evidence. In this section, we aim to explore the use of non-invasive respiratory support measures, ventilation strategies, and other advanced measures available to optimise patient care in the ICU.

### 4.1. High-Flow Nasal Oxygen (HFNO)

Asthma patients admitted to the ICU are often hypoxic, and supplemental oxygen therapy is one of the most common first-line treatments provided either via low-flow nasal cannula (LFNC) or via facemask [13]. LFNC can deliver supplemental oxygen at a maximum rate of 4 to 6 litres (L) per minute, while a standard reservoir mask can deliver a maximum rate of 15 L per minute [13]. Recently, several small studies have been performed evaluating the use of high-flow nasal oxygen (HFNO) for asthma exacerbation. A 2020 pilot randomised controlled trial of 37 patients compared HFNC therapy to conventional oxygen therapy in acute severe asthma patients and found that HFNC therapy reduced the severity of dyspnoea and respiratory rate in hypoxemic patients [52]. A 2021 study that randomised 62 acute severe asthma patients to either non-invasive ventilation (NIV) or HFNC found that the HFNC group had significantly lower respiratory rates at 48 h when compared with NIV [53]. Furthermore, the HFNC group had shorter mean ICU and hospital lengths of stay and a greater average comfort score [53]. A meta-analysis of four randomised controlled trials totalling 175 patients showed that although HFNC appeared to significantly lower dyspnoea score, there was no significant influence on gas exchange, intubation rate, or hospital length of stay [54]. Although a trial of HFNC could be considered for improvement of dyspnoea in certain patients, it should not be regarded as an alternative to mechanical ventilation.

### 4.2. Non-Invasive Ventilation (NIV)

Non-invasive ventilation (NIV) has the physiological effects of reducing the work of breathing and alveolar recruitment, improving dynamic lung compliance, offsetting the intrinsic positive end expiratory pressure (PEEP), and improving gas exchange during acute respiratory failure [55,56]. Intensive care admission and invasive mechanical ventilation (IMV) for acute respiratory failure from asthma are associated with significant morbidity. While NIV is an established practice for hypercapnic respiratory failure for COPD patients, the use of NIV during acute asthma exacerbations remains a contentious issue. Patients with acute severe asthma often present with hypoxemic respiratory failure rather than hypercapnic respiratory failure, and the primary aim is to optimise oxygenation while alleviating airway symptoms with bronchodilator therapy. However, patients with later-stage life-threatening or near-fatal asthma may develop hypercapnic respiratory failure, often necessitating endotracheal intubation and mechanical ventilation. Despite the lack of robust clinical evidence and guidance on using non-invasive ventilation in asthma, more recent retrospective cohort studies documented an increased frequency of NIV use [7,57–59]. However, NIV failure can be associated with an increased length of hospital stay and in-hospital mortality [60].

Although multiple randomised controlled trials have been published in this area, the studies consisted of small number of patients and were underpowered to assess robust clinical outcomes such as the need for IMV or mortality. A Cochrane review conducted over a decade ago summarised the findings from five small RCTs of 206 participants and concluded that there was insufficient clinical data to support the use of NIV in status asthmaticus [61]. However, more recently, a systematic review and meta-analysis of 10 pae-

diatric trials with 558 participants concluded that NIV use is associated with improved gas exchange, lower respiratory rate, and shorter hospital stay [62]. In a large multi-centre retrospective cohort study in ICUs, a quarter of patients received NIV for acute asthma exacerbation, and for those who received invasive mechanical ventilation (27%), 21% received NIV before IMV initiation. Moreover, the use of NIV also increased over the study period, from 18.5% in 2010 to 29.9% in 2017 [57]. However, a third of the studied patients had obesity (32.9%), and a quarter had obstructive sleep apnoea (26.7%), patient groups who are more likely to benefit from NIV [63]. Moreover, these cohort studies do not differentiate between patients with hypoxemic and hypercapnic respiratory failure. Large randomised controlled trials are required to assess the clinical efficacy of NIV in acute asthma exacerbations for preventing IMV and the impact on mortality. While NIV is not an alternative to IMV, the use of NIV should be limited to safe areas where IMV can be utilised rapidly, and at-risk patients can be monitored very closely.

### 4.3. Intubation

Approximately 2% of all acute severe asthma patients will go on to require intubation and mechanical ventilation [7]. Among ICU patients, nearly 36–46% may require invasive mechanical ventilation within 24 h of admission [6,64]. The decision to intubate is a clinical one, albeit considering other investigations and the projected deterioration of the patient's condition. Although variable depending on a combination of factors, the clinical indications for intubation in acute severe asthma are detailed below [65–67] (Table 1).

**Table 1.** Relative and immediate indications for intubation [65–67].

| Relative Indications | Immediate Indications |
| --- | --- |
| Progressive exhaustion | Cardiac arrest |
| Increasing use of accessory muscles or change in rate/depth of respiration | $PaO_2 < 8.0$ kPa and/or $PaCO_2 > 6.5$ kPa |
| Change in posture or speech | Severe obtundation or coma |
| Failure to reverse severe respiratory acidosis despite intensive therapy | Impending respiratory failure with gasping or inability to speak |
| Altered sensorium | Respiratory arrest |
| Severe hypoxemia with maximal oxygen delivery | |
| Silent chest | |

Intubation in the context of these relative indications can be challenging. The intubation of such patients requires rapid sequence induction with close monitoring of cardiovascular status. Preoxygenation may not be straightforward due to air trapping. A larger endotracheal tube is preferred, as it may help with subsequent bronchial toileting with the use of a bronchoscopy [68]. The induction agents commonly used are propofol and ketamine in combination with opiates and a paralysing agent. There are several potential complications that may arise during endotracheal intubation. Some are specific to asthma patients. These include worsening bronchospasm and significant air trapping during and after intubation [65]. Cardiovascular collapse from depleted intravascular volume status, vasodilation from anaesthetic medications, and increased intrathoracic pressure leading to reduced cardiac output need to be anticipated and managed effectively to prevent cardiac arrest [65,69,70] (Table 2).

**Table 2.** Potential complications arising from endotracheal intubation in an asthmatic patient [65,69,70].

| Potential Complications during Intubation |
| --- |
| Laryngospasm |
| Worsening bronchospasm |
| Significant air trapping |

**Table 2.** *Cont.*

| Potential Complications during Intubation |
|---|
| Aspiration |
| Barotrauma/volutrauma |
| Cardiovascular collapse |
| Cardiac arrhythmias |
| Cardiac arrest |

*4.4. Mechanical Ventilation*

Although life-threatening asthma is rare, a small proportion of patients may require IMV. While IMV is not a specific therapy for asthma, it is offered as a last resort while allowing time for bronchodilator therapy to take effect. Severe airflow limitation is the hallmark of acute asthma exacerbations. Patients with life-threatening asthma may present with severe respiratory distress, hypoxemia, arrhythmias, and pulse paradoxes due to air trapping with cardiovascular instability or, at later stages, hypercapnia and respiratory arrest. High airway pressures and dynamic hyperinflation can lead to air trapping, barotrauma, the development of pneumothorax, and hemodynamic instability [71]. The principle of mechanical ventilation in asthma is to avoid dynamic hyperinflation. The ventilation mode can be volume-assisted or controlled pressure-cycled ventilation with a tidal volume of 6–8 mL/kg IBW as a starting point. Minute ventilation should be adjusted to allow adequate time for the expiratory phase. Higher inspiratory to expiratory ratio (I:E) time is often required at 1:3 or 1:4 with a low respiratory rate (e.g., 12–14/min). In the event of continued breath stacking, disconnecting the ventilator circuit followed by manual decompression is required to improve the dynamic hyperinflation. Figure 2 is a pictorial depiction of the flow-volume loop during airflow obstruction and air trapping during acute asthma [72–74].

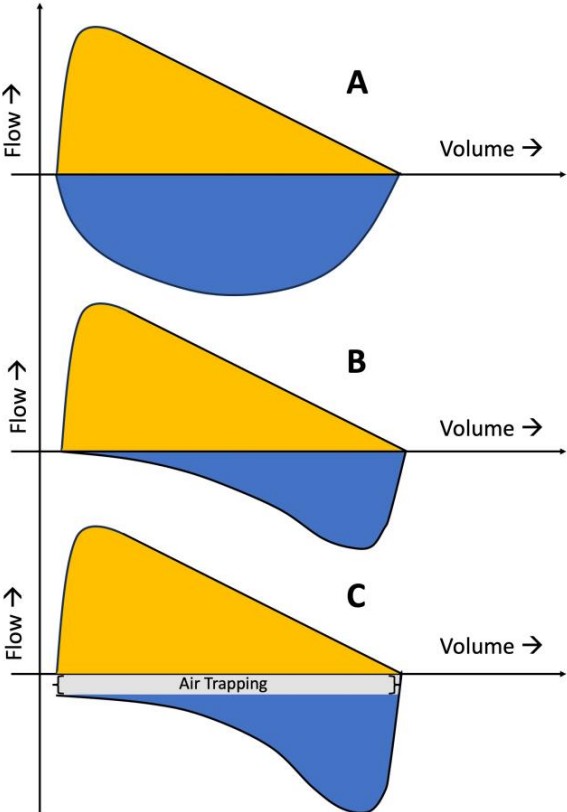

**Figure 2.** Flow volume loops depicting (**A**) normal pattern, (**B**) obstructive pattern, and (**C**) air trapping during severe asthma exacerbation. Yellow: Inspiration; Blue: Expiration and Grey: Air trapping.

Positive end-expiratory pressure (PEEP) is generally applied to all mechanically ventilated patients to avoid alveolar collapse, and higher levels of PEEP are often utilised for continued lung recruitment during mechanical ventilation in ARDS patients [75,76]. In patients with asthma, airway obstruction and expiratory flow limitation result in the generation of intrinsic PEEP, which can be measured by performing an expiratory breath hold on the ventilator. Excessive intrinsic PEEP can lead to gas trapping and, consequently, dynamic hyperinflation. Establishing optimal PEEP during mechanical ventilation for patients with asthma is a contentious issue. In practice, most advocate an applied PEEP of zero or reduced levels of PEEP below the intrinsic PEEP to minimise hyperinflation. Permissive hypercapnia is often adopted with deep sedation and neuromuscular paralysis at early stages to prevent patient–ventilator desynchrony. Standard ideal ventilator settings following mechanical ventilation are detailed in Figure 1. Reviews of mechanical ventilation strategies in asthma are described elsewhere by several in-depth reviews [71,77–79].

### 4.5. Anaesthetic Agents

#### 4.5.1. Ketamine

Ketamine is a dissociative anaesthetic used extensively for its sedative and analgesic effects. It is a non-competitive N-methyl-D-aspartate (NMDA) receptor antagonist, which is a derivative of phencyclidine. Its molecule contains an asymmetrical carbon atom with two enantiomers, S (+) ketamine and R (−) ketamine, and racemic preparations commonly contain equal concentrations of the enantiomers. It has excellent bioavailability from both intravenous and intramuscular administration with rapid onset of action and time-to-peak concentration of approximately 60 s. The duration of action of a single bolus is 10–15 min, and its distribution half-life is 7–11 min, being cleared via the hepatic route within 2–3 h [80]. The bronchodilator effect of ketamine was reported not long after its introduction to the market, with its use on children producing remarkable improvements in wheeze and respiratory distress [81]. The use of ketamine in severe refractory asthma has continued since, particularly in the paediatric population. Ketamine's effects appear to be mediated not just via NMDA receptor antagonism but via a myriad of other receptor targets, including dopaminergic, serotonergic, adrenergic, opioidergic, cholinergic, and ion channels, including voltage-gated sodium channels and hyperpolarisation-activated cyclic nucleotide gated channels [82].

The effect of bronchodilation is thought to be mediated not only by direct antagonism of NMDA receptor-induced bronchoconstriction but by an increase in synaptic catecholamine levels via presynaptic reuptake inhibition, downregulation of intrinsic nitric oxide synthetase activity (and therefore reduction in nitric oxide levels), airway smooth muscle relaxation via reduction of L-type calcium channel-induced calcium influx, and reduction of inflammation via reduction in macrophage recruitment [83–85]. Ketamine appears to be used more frequently in the paediatric population as an adjunct to standard therapy in refractory asthma exacerbations. Studies in this population, as in adults, have generally not borne out conclusive benefits compared to standard therapy. Few small randomised controlled trials in the adult population have evaluated the use of ketamine in patients with asthma and reported outcomes on respiratory mechanics and gas exchange variables [86,87]. Another randomised, double-blind, placebo-controlled trial showed no increased bronchodilatory effect when given at a dose low enough to avoid dysphoric reactions when compared with standard therapy for treating asthma exacerbations in the ED [88]. A systematic review of seven prospective studies (including two paediatric studies) concluded that there was no clear benefit from the administration of IV ketamine in patients with refractory asthma, and further multi-centre randomised controlled trials are required to conclusively evaluate this finding [83]. However, ketamine is commonly used as an induction and maintenance of sedation agent in patients with severe asthma [89,90]. In summary, it appears that although there is a potentially physiological argument for the use of ketamine as an adjunct to improve bronchoconstriction, and anecdotal data appear to

suggest some benefit, the literature thus far does not support its use. Larger, high-quality trials are likely required to explore this further.

### 4.5.2. Inhalational Anaesthetic Agents

Inhalational anaesthetic agents such as isoflurane, sevoflurane, enflurane, and halothane are known to induce bronchodilation through the blockage of voltage-gated calcium channels, depletion of sarcoplasmic reticulum calcium stores, and possibly through GABAergic mechanisms; however, this effect is not seen with desflurane [91–93]. There was early evidence in the 1980s and 1990s of a clinical benefit in treating severe refractory asthma with halothane or isoflurane in both adults and children; however, these studies were small, involving only a few patients [94,95]. Isoflurane improves arterial pH and reduces partial pressure of arterial carbon dioxide in mechanically ventilated children, and sevoflurane may provide clinical improvement in mechanically ventilated children with life-threatening asthma [96,97]. Further smaller studies and case reports have continued to show clinically beneficial bronchodilation from volatile agent administration, but to date, no randomised, placebo-controlled trials exist to explore these benefits conclusively [98–100].

There is sufficient theoretical and clinical evidence to suggest that a trial of volatile anaesthetics may be beneficial in severe refractory asthma exacerbations. Indeed, BTS guidelines suggest that a trial of sevoflurane could be beneficial where appropriate gas scavenging facilities exist [10]. It must be noted that use of inhalational anaesthetic agents in the ICU may only be performed in the presence of appropriate gas-scavenging equipment along with specialised delivery devices which are designed to conserve inhalational anaesthetics. One such device is the AnaConDa™, which enables the administration of volatile anaesthetics (isoflurane or sevoflurane) and is inserted between the Y-piece and ET tube. While its use as an additional rescue therapy continues, further studies are needed to evaluate the clinical impact of such devices on robust patient outcomes [101].

### 4.6. Extracorporeal $CO_2$ Removal (ECCO$_2$R)

Extracorporeal carbon dioxide removal (ECCO$_2$R) provides a means of fast and effective treatment of $CO_2$ removal in acute hypercapnic respiratory failure; however, in comparison to extracorporeal membrane oxygenation (ECMO), the blood is not oxygenated before returning to the body [102,103]. Given the solubility and diffusion characteristics of carbon dioxide compared to oxygen, ECCO$_2$R requires much lower blood flow rates. Utilising this system, the deleterious effects of IMV can be lessened [104,105]. Several case reports have demonstrated the efficacy of ECCO$_2$R in the adult asthma population [106–108]; however, only limited comparative robust data exist on this topic. Bromberger et al. retrospectively analysed 26 adult patients from a single centre who required ECCO$_2$R for status asthmatics, demonstrating 100% survival to hospital discharge with significant improvements in tidal volumes, respiratory rate, intrinsic PEEP, peak and plateau pressures, and acidosis following one day of ECCO$_2$R [109]. Additionally, the use of vasopressors was significantly reduced after ECCO$_2$R initiation, highlighting that correction of acidosis with subsequent reduction in ventilator pressures can improve hemodynamic stability. Additionally, 76.9% of patients were able to be extubated whilst ECCO$_2$R was being delivered, further reducing the incidence of ventilator-associated lung injuries commonly seen in asthma [109].

The REST randomised trial investigated the use of ECCO$_2$R in acute hypoxic respiratory failure. In this trial, 412 patients were randomised to ECCO$_2$R with reduced tidal volume ventilation or standard intensive therapy. There was no difference in 90-day mortality; however, ECCO$_2$R was associated with significantly reduced ventilator-free days and a much higher incidence of serious adverse complications (31% versus 9%), including intracranial haemorrhage. This trial was subsequently terminated early, and therefore, despite these concerning initial results, the study may have been too underpowered to show any potential benefit [110]. More research is required in this area, and in the UK, NICE guidelines support the use of ECCO$_2$R only in the context of a clinical trial [111].

### 4.7. ECMO

Extracorporeal membrane oxygenation (ECMO) is an invasive supportive treatment used in critical care for several respiratory pathologies refractory to standard intensive clinical practices (including pneumonia, ARDS, pulmonary embolism, and trauma) [112]. The various mechanisms by which ECMO can provide organ support depend upon its configuration. Blood extracted from a central vein passes through an oxygenator, which extracts carbon dioxide whilst saturating the blood with oxygen. The blood then passes back to the body via the same or different central vein (V-V-ECMO). V-V ECMO provides respiratory support by essentially bypassing the lungs; however, it necessitates a stable functioning circulatory system. An alternative configuration is to return the blood at higher pressures directly to the arterial circulation (V-A-ECMO), thus bypassing both the heart and lungs, providing cardiovascular and respiratory support [113,114].

The physiological benefits of ECMO during acute respiratory failure are to provide sufficient gas exchange whilst preventing ventilator-induced lung injury, reduce metabolic demand of breathing, and provide a means of lung "ultra-protection" with the use of low tidal volumes and prevention of atelectasis and barotrauma [115]. Furthermore, patients on ECMO can better tolerate more invasive airway clearance manoeuvres, such as bronchoscopy and bronchial washes, reducing the incidence of mucus plugging [116].

Following the CESAR trial in 2009 and the recent COVID-19 pandemic, the global utilisation of ECMO has expanded greatly, and in the UK, ECMO is only delivered by dedicated specialist centres. The CESAR results demonstrated a relative risk of survival to 6 months without disability when ECMO was used on adults with severe, potentially reversible respiratory failure compared to standard therapy alone [117]. An important confounder in this study was that there was demonstrable improved success in patients admitted directly to ECMO centres versus those who had to be transferred to them, highlighting the importance of earlier intervention for those who require it. Delivery without complications, however, require both specialist equipment and staff; therefore, expansion without appreciation for this may in fact worsen outcomes [118,119].

In asthma refractory to standard intensive therapy (near-fatal asthma or status asthmaticus), ECMO may help to improve gas exchange without the expense of aggressive IMV. Data from the extra corporeal life support organisation registry from 1992 to 2016 demonstrated that in 272 patients with life threatening asthma, the survival rate to hospital discharge was 83.5%, with improvements in the fraction of inspired oxygen, peak inspiratory pressures, driving pressures, and mean airway pressure following the initiation of ECMO [120]. Additionally, an analysis of 1205 patients in the UK who required ECMO demonstrated an overall survival rate 74%, but near-fatal asthma as an indication for ECMO (10%) was associated with a survival rate of 95% compared to 71% for other respiratory pathologies. Additionally, younger patients and those with better oxygenation at the point of ECMO initiation were more likely to survive [121].

The most frequently observed complications of ECMO are mainly related to cannulation and include bleeding/thrombosis and infection. The rate of complication determined by Yeo et al. (2017) was 65.1%, with haemorrhage at 28.3%, but this resulted in very few deaths; however, bleeding was an independent risk factor for in-hospital mortality [120]. More conservative anticoagulation may reduce the incidence of this complication [122]. However, treading the line between bleeding and clotting is difficult in critical care, and the use of ECMO can make this even more challenging. Clinicians must evaluate clotting/bleeding risk both at initiation and throughout ECMO delivery.

Despite the survival benefit demonstrated above and in numerous case reports [123–126], high-quality comparative data to support ECMO use is limited. This owes to the rarity of ECMO use in near-fatal asthma and that it is not a standard feature in asthma guidelines [10]. In addition, the evidence determined from the ECMO registry lacks selection criteria, does not report on whether patients were on appropriate/optimal ventilator settings prior to ECMO initiation, and is reliant on voluntary data reporting, risking reporting bias [10].

### 4.8. Mucolytics

Since one of the primary pathophysiological mechanisms of severe asthma exacerbations and mortality is inflammation with a subsequent increase in airway secretions and cast formation, it is thought that the management of mucous plugging could lead to possible improvement in outcomes. A range of mucolytic therapies have been trialled to combat this phenomenon, with varying efficacy.

#### 4.8.1. Nebulised Heparin

Heparin is an endogenous glycosaminoglycan used primarily for its anticoagulant properties. Inhaled heparin has previously been proposed as an adjunct to treat severe asthma due to speculation regarding its possible anti-inflammatory effects. Heparin has been shown to bind and inhibit a variety of cytotoxic and inflammatory mediators, including inhibition of mast cell activation and mast cell mediators along with inhibition of eosinophil cell migration and cationic proteins [127]. Inhalation of heparin is likely to suppress the initial reaction to allergens and exercise-induced asthma by preventing the release of mast cell mediators [127,128]. Although several early studies in the 1960s and later in the 1990s reported subjective improvement in asthma symptoms, objective evidence for clinical improvement has not been demonstrated [129,130]. Two case studies have been reported showing a benefit of nebulised heparin administered early in the course of asthma exacerbations; however, little can be gleaned from these in terms of wider clinical effects [127]. A meta-analysis published in 2023 examined 23 studies in their initial qualitative analyses, reducing this to eight studies (220 patients) for the primary outcome measure of improvement in respiratory function (FEV1% and FEV1). A statistically significant improvement in FEV1 suggesting that a trial of inhaled heparin may prove beneficial in addition to standard therapy was found. Interestingly, upon subgroup analysis, it was found that unfractionated heparin was more effective than low-molecular-weight heparin for this indication [128]. Further large randomised controlled trials are needed to evaluate the prophylactic and therapeutic effect of nebulised heparin in mechanically ventilated asthmatic patients.

#### 4.8.2. Recombinant Human Deoxyribonuclease (rhDNase/Dornase Alfa)

Airway obstruction by viscous mucus is one of the primary pathophysiological features of fatal asthma exacerbations. Recombinant human deoxyribonuclease (rhDNase) has been used to treat several respiratory conditions, most notably cystic fibrosis, for many years. The lysis of inflammatory cells leads to free DNA that is thought to be a primary contributor to the viscosity and adhesiveness of mucus. rhDNase is reported to liquefy these components by depolymerising extracellular DNA and thus reducing its size, leading to transformation of viscous mucus into a flowing liquid [131,132]. rhDNase is currently only licensed for use in cystic fibrosis as a solution of 2500 units delivered up to twice per day via a jet nebuliser. A small study of 50 patients with symptomatic asthma in the emergency department did not show any clinical improvement [133]. The British Thoracic Society guidelines on the management of asthma do not support its use due to a lack of sufficient evidence [10]. Much of the literature involving rhDNase is focused on treating patients with cystic fibrosis, with only a minority of studies looking at its use in asthma. Case studies dating back to the 1990s have reported excellent effects with its use in specific patients who had proven atelectasis or lobar mucus plugging [134]. However, case reports suggest potential beneficial effects in mechanically ventilated asthmatic patients, and further studies are needed to evaluate the potential clinical benefits of both nebulised and bronchoscopic administration of rhDNase on clinical outcomes [135,136].

### 4.9. Heliox

The primary components of atmospheric air are nitrogen (~78%) and oxygen (~21%). A mixture of helium (He) and oxygen ($O_2$) (known colloquially as heliox) has been used to reduce airway resistance and thereby reduce the work of breathing. While having a

similar viscosity to nitrogen, helium has a higher thermal conductivity, and therefore, a heliox gas mixture of 79% helium and 21% oxygen has a density almost six times lower than atmospheric air. Further, $CO_2$ diffusion and elimination appear to be increased since it diffuses four to five times faster through a mixture of He-$O_2$ than through $N_2$-$O_2$. Finally, helium mixtures have been shown to mediate redistribution of gas throughout the lungs and produce an increase in alveolar ventilation through a reduction in $CO_2$ dead space, facilitated by a better diffusion of $CO_2$ in the low-density helium-oxygen mixture [137,138].

Helium mixtures were used as early as the 1930s in respiratory distress, since little else was available for the treatment of acute bronchospasm. It fell out of favour since supplies dwindled during World War II, and the subsequent invention of dedicated potent bronchodilators made it effectively obsolete for this purpose [137]. There are controversies over its use due to potentially consequently lower oxygen concentrations used in the gaseous mixture and due to its relatively complicated technical application and high costs. The flow meters and nebuliser generator systems used in ventilated patients must be adapted to use heliox [139]. A 2006 meta-analysis of ten trials containing 544 acute asthma patients studied the effect of heliox administration on pulmonary function tests for non-intubated acute asthma patients. Heliox only improved pulmonary function in a subgroup of patients with the most severe baseline impairment; however, this conclusion was based on a small number of studies. The analysis included trials from both adults and children, and there were no significant differences between these groups [140]. A further 2014 meta-analysis of 11 trials and 697 patients showed an overall statistically significant increase in mean peak expiratory flow. Furthermore, the effect was maximised in patients suffering the most severe asthma exacerbations [141]. There is ample evidence expounding the theoretical benefits of helium, with some small studies and meta-analyses showing clinical benefit; however, there do not exist any high-powered studies showing significant clinical benefit [138,139,142]. Again, similar to other ICU interventions, further studies are needed to evaluate the use of heliox in patients with acute asthma exacerbation in the ICU.

Limitations of This Review

We would like to acknowledge that this narrative review has some limitations due to methodological constraints and significant heterogeneity in the studies included, such as the type of study, population studied, and interventions provided. Nevertheless, we have identified significant gaps in the evidence relating to the management of patients with life-threatening and near-fatal asthma exacerbations in the ICU setting.

## 5. Conclusions

Managing life-threatening and near-fatal asthma exacerbations in ICU can be challenging. Although various interventions are often used in combination in extreme cases, such as intravenous bronchodilators, non-invasive ventilation, intravenous ketamine, volatile anaesthetic agents, specific mucolytics, $ECCO_2$ removal devices, and ECMO, there is little clinical evidence to support their effectiveness. Therefore, guidelines for managing such patients in the ICU need to specifically address this significant research gap. An evidence-based approach is necessary to evaluate these interventions beyond the standard guidelines. We urge major respiratory and intensive care societies to consider this cohort as an orphan disease and address these urgent research needs.

**Author Contributions:** Conceptualisation, T.T., T.R. and A.D.; methodology, T.T., T.R. and A.D.; writing—original draft preparation, T.T., T.R. and A.D.; writing—review and editing, T.T., T.R. and A.D.; visualisation, T.R. All authors have read and agreed to the published version of the manuscript.

**Funding:** This research received no external funding.

**Institutional Review Board Statement:** Not applicable.

**Informed Consent Statement:** Not applicable.

**Data Availability Statement:** Not applicable.

**Conflicts of Interest:** The authors declare no conflicts of interest.

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
