# Peer review of "Management of Acute Life-Threatening Asthma Exacerbations in the Intensive Care Unit"

_applsci, doi:10.3390/app14020693_

Round 1

Reviewer 1 Report

Comments and Suggestions for Authors

The manuscript by Talbot et al. provided a very interesting and comprehensive analysis on the management of acute forms of asthma in a critical setting. The topic is worthy of attention. The review is sound and well written. I have only minor issues to report:

- Line 29. Please provide a reference for this sentence on the epidemiology and severity of asthma.

- Figure 1. Please provide an explanation for the different colors used in the figure. Is it related to the amount of available evidence? 

- Line 129. Please explain the possible reason for the advantages of early administration.

- Line 169. I believe this paragraph should be slightly summarized by removing the oldest references and focusing on the most recent available literature.

Author Response

We are very grateful for the reviewer's comments and please see our response to the comments below;

The manuscript by Talbot et al. provided a very interesting and comprehensive analysis on the management of acute forms of asthma in a critical setting. The topic is worthy of attention. The review is sound and well written. I have only minor issues to report:

- Line 29. Please provide a reference for this sentence on the epidemiology and severity of asthma.

Response: We have provided a reference for this sentence.  

- Figure 1. Please provide an explanation for the different colors used in the figure. Is it related to the amount of available evidence? 

Response: We have now clarified in the figure legend section.

- Line 129. Please explain the possible reason for the advantages of early administration.

Response: We have modified the sentence to explain the reason for early administration.

- Line 169. I believe this paragraph should be slightly summarized by removing the oldest references and focusing on the most recent available literature.

Response: We have modified this paragraph and removed any inference to old literature

Reviewer 2 Report

Comments and Suggestions for Authors

Abstract.

#1 Please avoid the use of non-standard abbreviations such as “NIV” in the Abstract.

Introduction.

#2 Page one, line 41-42: Please remain consistent regarding the number of decimal places when giving percentages.

#3 Page one, line 44: References 2, 3, 5 are listed twice.

Main part.

#4 I appreciate that the authors summarized existing evidence in Figure 1. There are some options for improvement of this figure. It is unclear what the y-axis shows – are the options ordered towards escalation of therapy? What do the colors of the different boxes signify? Please also explain non-standard abbreviations used in the figure.

#5 The part on management of acute asthma exacerbations would benefit from a short paragraph on non-pharmacological interventions.

#6 Page three, lines 114-115: “…for example salbutamol directly activates the adrenoceptor while salbutamol interacts with a receptor-specific auxiliary binding site [16]” seems off – salmeterol, not salbutamol interacts with a receptor-specific auxiliary binding site, according to the provided reference.

#7 In the section on nebulized bronchodilators, please kindly also mention side effects, which are of relevance for patients in intensive care settings. Did previous studies evaluate these effects in this cohort?

#8 Please avoid verbiage such as “The study was unable to demonstrate…” (page four, line 155).

#9 A variety of statements stand unreferenced, which is not acceptable. Please kindly add references to support these. Just some examples: Page nine, lines 392-393; page twelve, lines 518-521; page six, lines 241-245.

#10 Page six, lines 239-240: Spacing inconsistency.

#11 Page seven, Table 2: Please clarify whether these complications associated with endotracheal intubation itself, or with intubation in the setting of acute asthma exacerbation. Please add respective references and provide effect estimates of selected studies to help quantify the associated risk in Table 2.

#12 Page nine, lines 394-402: Please add a justification or references for these ventilator regimen recommendations. The review should also present different studies on these ventilator settings who contributed to generating these recommendations.

#13 Page nine, line 406: Reference is entirely missing.

#14 The section about Heliox for acute asthma exacerbations is very interesting, however the mentioned studies are more than ten years old. A short literature search revealed that more recent studies are available, which should be included to bring an aspect of novelty.

#15 Generally, please add more effect size measures when commenting on the efficacy of interventions shown in studies. This helps understand the magnitude of effect in the context of the respective study design and setting.

Author Response

We are grateful for the reviewer's comments, and please see the responses below. 

Abstract.

#1 Please avoid the use of non-standard abbreviations such as “NIV” in the Abstract.

Response: We have now removed these abbreviations.

Introduction.

#2 Page one, line 41-42: Please remain consistent regarding the number of decimal places when giving percentages.

Response: We have modified this as suggested. 

#3 Page one, line 44: References 2, 3, 5 are listed twice.

Response: We have removed these now.

Main part.

#4 I appreciate that the authors summarized existing evidence in Figure 1. There are some options for improvement of this figure. It is unclear what the y-axis shows – are the options ordered towards escalation of therapy? What do the colors of the different boxes signify? Please also explain non-standard abbreviations used in the figure.

Response: We have modified the figure and amended the figure legend to clarify this further.

#5 The part on the management of acute asthma exacerbations would benefit from a short paragraph on non-pharmacological interventions.

Response: While we agree there are several non-pharmacological measures which are helpful for asthma and acute asthma. This is not relevant in an ICU setting during life-threatening exacerbation.

#6 Page three, lines 114-115: “…for example salbutamol directly activates the adrenoceptor while salbutamol interacts with a receptor-specific auxiliary binding site [16]” seems off – salmeterol, not salbutamol interacts with a receptor-specific auxiliary binding site, according to the provided reference.

Response: This was a typo, and we have corrected it now (line 166).

#7 In the section on nebulized bronchodilators, please kindly also mention side effects, which are of relevance for patients in intensive care settings. Did previous studies evaluate these effects in this cohort?

Response: We have added a sentence to include the side effects (line 199-202).

#8 Please avoid verbiage such as “The study was unable to demonstrate…” (page four, line 155).

Response: We have modified this sentence now (line 221-222).

#9 A variety of statements stand unreferenced, which is not acceptable. Please kindly add references to support these. Just some examples: Page nine, lines 392-393; page twelve, lines 518-521; page six, lines 241-245.

Response: We have provided references to these statements now.

#10 Page six, lines 239-240: Spacing inconsistency.

Response: We have corrected it as suggested.

#11 Page seven, Table 2: Please clarify whether these complications associated with endotracheal intubation itself, or with intubation in the setting of acute asthma exacerbation. Please add respective references and provide effect estimates of selected studies to help quantify the associated risk in Table 2.

Response: These complications are generic and specific to asthma. We have clarified these. There are no large studies to quantify these, as nearly all patients will have airway-related complications. We are not able to provide absolute risks associated with intubation specific to asthma as no large data set exists. Translating effect estimate data from the generic ICU population is misleading.

#12 Page nine, lines 394-402: Please add a justification or references for these ventilator regimen recommendations. The review should also present different studies on these ventilator settings that contributed to generating these recommendations.

Response: Most of these come from very old studies, clinical practice experience, and review articles. We have now provided additional references. 

#13 Page nine, line 406: Reference is entirely missing.

Response: We have provided a reference for this sentence now.

#14 The section about Heliox for acute asthma exacerbations is very interesting, however, the mentioned studies are more than ten years old. A short literature search revealed that more recent studies are available, which should be included to bring an aspect of novelty.

Response: These are old interventions, and emerging evidence is lacking due to the lack of new studies. We have added a more recent systematic review. 

#15 Generally, please add more effect size measures when commenting on the efficacy of interventions shown in studies. This helps understand the magnitude of effect in the context of the respective study design and setting.

Response: While we agree with the reviewer, most studies have no clinical effect or report physiological outcomes without robust clinical outcomes such as mortality or the need for mechanical ventilation. Consequently, reporting the magnitude of these simple physiological measures and their effect has no clinical value.

Reviewer 3 Report

Comments and Suggestions for Authors

Dear Dr. Jantamai,

Thank you for the opportunity to review the manuscript entitled “Management of acute life-threatening asthma in ICU” worth being published in the special issue Asthma and Respiratory Disease: Prediction, Diagnosis and Treatment, Volume II in Applied Sciences following major revision.

The authors present an interesting narrative review that encompasses more than 100 important articles on asthma in critically ill patients. The authors provide a thorough overview of available treatments and potential future directions in the management of acute asthma exacerbations in the intensive care unit (ICU). Based on their findings, the authors highlight the need for multicenter trials to create evidence for specific guidelines that currently do not yet exist. The review is well-written, and the main messages the authors try to bring to the audience are clear. I commend the authors for their excellent work and recommend the acceptance of this review for publication following a major revision. Predominantly, the authors need to be more specific when discussing current literature. While some of the literature the authors present is fairly old (>10 years old), they clearly need to provide details on the results. Please see my detailed comments below.

Title

#1 Please refer to acute life-threatening asthma exacerbations in the intensive care unit.

Abstract

#2 Please introduce abbreviations such as NIV or ICU.

Introduction

#3 The introduction should highlight the relevance of asthma to the general population. The authors suggested to provide specific numbers on a) the estimated number of patients suffering from asthma (either worldwide or in England, which would be in line with l. 29-34) and b) the economic consequences of asthma (if possible).

#4 It would be great if the authors could state how many patients with asthma have a once-in-a-lifetime asthma exacerbation.

#5 2.36/100,000 should just be stated as the % value. Please adhere to a consistent display of decimals (either X.X or X.XX)

#6 Please avoid the word “asthmatics” and instead refer to “patients with asthma”.

#7 Please introduce the abbreviation UK in l. 46 and then use it consistently throughout.

#8 p. 2, l. 53 – It would be helpful if the authors cite the guidelines on asthma management in this sentence.

#9 p. 2, l. 58 – Please remove the sentence on the pediatric population from the introduction and implement it at a later place in the discussion of the review.

Classifications of asthma severity

#10 The figure is interesting to the audience. The authors may want to modify the y-axis as it is not entirely clear that a greater y-value shows more invasive therapeutic options. It is further unclear what the authors are trying to show with the color coding – is it the level of evidence?

#11 The abbreviation USA needs to be introduced.

#12 The text says that “Figure 1 is a summary of the clinical features” – I would disagree with that as only the x-axis shows the clinical features – however; in my opinion the focus of Figure 1 lies in the different treatment options dependent on the clinical features. This should be the main message of figure 1. I would further appreciate it if the authors provide a more detailed figure legend.

Management of acute asthma exacerbations

#13 I am curious whether the authors considered the discussion of using nitrous oxide in patients with asthma.

#14 p. 3, l. 93 – Please provide details on the adverse outcomes investigated in the IOTA trial.

#15 The authors mention there is conflicting evidence – Can the authors perhaps help to explain why there is conflicting evidence? Are there differences in the trials with regard to patients, endpoints, and methods? This would be very important to mention in this review.

#16 As this review should be targeted at clinicians who may lack time to work through the entire BTS guideline, please state the currently recommended targets for supplemental oxygen.

#17 The authors should clearly state which nebulized bronchodilator (salbutamol?) is standard practice.

#18 p. 3, l. 125 – A Cochrane review from 2001 is too old for a review written in 2023. The authors should focus on the most relevant literature within the past 5 (to a maximum of 10 years).

#19 In critically ill patients, steroids have been associated with adverse outcomes. The authors should implement the discussion of the benefits vs. harm of glucocorticoids in critically ill patients with asthma exacerbations.

#20 Great paragraph on Magnesium sulfate – excellent work.

#21 p. 5, l. 189 – Please revise the sentence on “Adult randomized controlled trials”. This sentence should be “Randomized controlled trials in ### adults”

#22 p. 5, l. 223 – The abbreviation RCT needs to be introduced and then used consistently (already much earlier)

Acute-life-threatening asthma- advanced management in the ICU

#23 p. 5, l. 234 – Please be consistent in the use of abbreviations (here for example: ICU instead of intensive care unit).

#24 I suggest that the authors provide a helpful guide for decision-making to the authors on how ventilation should be managed. From my understanding, it should be 1. LFNC -> 2. HFNO -> 3. NIV and 4. Invasive ventilation. Showing this strategy for increasing treatment levels would be appreciated. The authors should also describe the side effects of the suggested options. The authors could further expand Table 1 and describe indications for each treatment level and when it should proceed to the next level. The same applies to Table 2.

#25 It is unclear why the paragraph on mechanical ventilation does not immediately follow the paragraph on intubation. The one is inevitably connected to the other….

#26 Again, I am missing the discussion of “inhaled” NO.

#27 Again, you often mention literature that is more than 10 years old (e.g., Heliox) – this is not acceptable in a world where research is evolving in an increasing pace like nowadays.

#28 As a general concern and suggestion for the discussion of studies – Please be more specific. Avoid saying unspecific results like “was associated with” but rather mention the detailed effect size. This comment applies to many locations in your review and is my major concern.

#29 I find Figure 2 not very helpful as it is very much common knowledge and does not add to this review. The authors may consider removing the figure and might rather provide a more extensive Table 1.

#30 For all strategies the authors are discussing, a systematic display of studies (e.g. a large table) might be helpful. I would appreciate if the authors could provide such a table that provides key facts in a more systematic fashion. Especially when the evidence is limited it might be helpful to see for the readers that there are just 1-2 studies showing that x is not very much backed up by literature in a table.

Conclusions

#31 Before the conclusion section, I would appreciate if the authors could state limitations of their review. Limitations may include literature access (I assume MEDLINE was used for the search?), the non-systematic review, as well as difficulty to compare studies because of large heterogeneity across the study groups.

#32 As the authors clearly suggest future studies, I want to challenge the authors on the following: It would be very helpful (but not mandatory) to provide a design for a randomized controlled trial. The authors may select a treatment option that they think is most urgent to be better understood (and perhaps most promising) and should provide some recommendations on the setting, the population as well as the definitions of exposure and endpoints.

Again, I thank the authors for this interesting and well-written review and look forward to receiving a revised version. Thank you very much. 

Author Response

We are really grateful for the reviewer's comments; please see the response. 

Thank you for the opportunity to review the manuscript entitled “Management of acute life-threatening asthma in ICU” worth being published in the special issue Asthma and Respiratory Disease: Prediction, Diagnosis and Treatment, Volume II in Applied Sciences following major revision.

The authors present an interesting narrative review that encompasses more than 100 important articles on asthma in critically ill patients. The authors provide a thorough overview of available treatments and potential future directions in the management of acute asthma exacerbations in the intensive care unit (ICU). Based on their findings, the authors highlight the need for multicenter trials to create evidence for specific guidelines that currently do not yet exist. The review is well-written, and the main messages the authors try to bring to the audience are clear. I commend the authors for their excellent work and recommend the acceptance of this review for publication following a major revision. Predominantly, the authors need to be more specific when discussing current literature. While some of the literature the authors present is fairly old (>10 years old), they clearly need to provide details on the results. Please see my detailed comments below.

Title

#1 Please refer to acute life-threatening asthma exacerbations in the intensive care unit.

Response: We have modified the title as suggested.

Abstract

#2 Please introduce abbreviations such as NIV or ICU.

Response: We have removed all the abbreviations from the abstract.

Introduction

#3 The introduction should highlight the relevance of asthma to the general population. The authors suggested to provide specific numbers on a) the estimated number of patients suffering from asthma (either worldwide or in England, which would be in line with l. 29-34) and b) the economic consequences of asthma (if possible).

Response: We have now introduced a sentence to highlight the global burden and economic consequence (line 27-30).

#4 It would be great if the authors could state how many patients with asthma have a once-in-a-lifetime asthma exacerbation.

Response: This data is not straightforward. There are specific risk factors including the type of asthma, number of previous exacerbations, compliance with medications, individual risk factors such as psychological problems, obesity, smoking and socioeconomic status all contribute to exacerbations. The data for acute severe exacerbation for high-risk, poorly controlled asthma has been published widely but varies across nations. Consequently, we have not presented it here.

#5 2.36/100,000 should just be stated as the % value. Please adhere to a consistent display of decimals (either X.X or X.XX)

Response: The data is adjusted as a number per 100,000 population. This is how the public health data is presented. We have now modified the sentence slightly to explicitly mention per 100,000 population. 

#6 Please avoid the word “asthmatics” and instead refer to “patients with asthma”.

Response: We have modified this throughout the manuscript as suggested.

#7 Please introduce the abbreviation UK in l. 46 and then use it consistently throughout.

Response: We have modified this throughout the manuscript as suggested.

#8 p. 2, l. 53 – It would be helpful if the authors cite the guidelines on asthma management in this sentence.

Response: We have inserted the references here.

#9 p. 2, l. 58 – Please remove the sentence on the pediatric population from the introduction and implement it at a later place in the discussion of the review.

Response: We have removed this sentence as suggested.

Classifications of asthma severity

#10 The figure is interesting to the audience. The authors may want to modify the y-axis as it is not entirely clear that a greater y-value shows more invasive therapeutic options. It is further unclear what the authors are trying to show with the color coding – is it the level of evidence?

Response: We have modified the figure and provided more clarification in the legend.

#11 The abbreviation USA needs to be introduced.

Response: We have now introduced an earlier sentence with the abbreviation.

#12 The text says that “Figure 1 is a summary of the clinical features” – I would disagree with that as only the x-axis shows the clinical features – however; in my opinion the focus of Figure 1 lies in the different treatment options dependent on the clinical features. This should be the main message of figure 1. I would further appreciate it if the authors provide a more detailed figure legend.

Response: We have modified this sentence and expanded the figure legend to clarify this further.

Management of acute asthma exacerbations

#13 I am curious whether the authors considered the discussion of using nitrous oxide in patients with asthma.

Response: While exhaled fractional nitric oxide (FeNO) is used as a marker of airway inflammation, the purpose of NO as a smooth muscle relaxant is not utilised in the ICU population. Nitrous oxide (N2O) is a weak anaesthetic not used for the treatment of asthma.

#14 p. 3, l. 93 – Please provide details on the adverse outcomes investigated in the IOTA trial.

Response: IOTA was a systematic review and meta-analysis and we have now added a sentence to clarify the increased risk of mortality with hyperoxemia (line 144-145).

#15 The authors mention there is conflicting evidence – Can the authors perhaps help to explain why there is conflicting evidence? Are there differences in the trials with regard to patients, endpoints, and methods? This would be very important to mention in this review.

Response: There are several ICU studies in ICU with varying population groups, interventions, targets and outcomes. There is a wealth of information from detailed review articles. Although none are specific to asthma, we have highlighted a few in the references (17-20).

#16 As this review should be targeted at clinicians who may lack time to work through the entire BTS guideline, please state the currently recommended targets for supplemental oxygen.

Response: We have modified the sentence to provide this target value (line 157-158).

#17 The authors should clearly state which nebulized bronchodilator (salbutamol?) is standard practice.

Response: We have mentioned this now.

#18 p. 3, l. 125 – A Cochrane review from 2001 is too old for a review written in 2023. The authors should focus on the most relevant literature within the past 5 (to a maximum of 10 years).

Response: Corticosteroids are an established treatment for acute asthma. While we agree with the reviewer that emerging evidence is important to review, there are unlikely to be any new studies investigating corticosteroids versus placebo for asthma exacerbation. 

#19 In critically ill patients, steroids have been associated with adverse outcomes. The authors should implement the discussion of the benefits vs. harm of glucocorticoids in critically ill patients with asthma exacerbations.

Response: We have introduced a sentence to address this comment (209-212).

#20 Great paragraph on Magnesium sulfate – excellent work.

Response: Thank you.

#21 p. 5, l. 189 – Please revise the sentence on “Adult randomized controlled trials”. This sentence should be “Randomized controlled trials in ### adults”

Response: We have modified the sentence as suggested.

#22 p. 5, l. 223 – The abbreviation RCT needs to be introduced and then used consistently (already much earlier)

Response: We have now clarified this in the first introduction of RCT.

Acute-life-threatening asthma- advanced management in the ICU

#23 p. 5, l. 234 – Please be consistent in the use of abbreviations (here for example: ICU instead of intensive care unit).

Response: We have amended these as suggested.

#24 I suggest that the authors provide a helpful guide for decision-making to the authors on how ventilation should be managed. From my understanding, it should be 1. LFNC -> 2. HFNO -> 3. NIV and 4. Invasive ventilation. Showing this strategy for increasing treatment levels would be appreciated. The authors should also describe the side effects of the suggested options. The authors could further expand Table 1 and describe indications for each treatment level and when it should proceed to the next level. The same applies to Table 2.

Response: While we agree with the reviewer that a treatment ladder for varying severity would be useful, this is not the case for most patients. This will depend on the tolerability of HFNO/NIV/VPAP, the requirement of pressure support, the degree of ventilatory failure and hypoxemia, the work of breathing and the availability of devices. Often, this will be a trial of treatment, and no head-to-head studies suggest this escalating management strategy.  

#25 It is unclear why the paragraph on mechanical ventilation does not immediately follow the paragraph on intubation. The one is inevitably connected to the other….

Response: We have modified this section to follow on from the intubation section as suggested

#26 Again, I am missing the discussion of “inhaled” NO.

Response: We have clarified this in the previous comment 13.

#27 Again, you often mention literature that is more than 10 years old (e.g., Heliox) – this is not acceptable in a world where research is evolving in an increasing pace like nowadays.

Response: We have modified this section now to include more recent literature (line 850-859).

#28 As a general concern and suggestion for the discussion of studies – Please be more specific. Avoid saying unspecific results like “was associated with” but rather mention the detailed effect size. This comment applies to many locations in your review and is my major concern.

Response: This manuscript aims to provide a narrative review of evidence rather than a detailed statistical systematic review of measures and effects.

#29 I find Figure 2 not very helpful as it is very much common knowledge and does not add to this review. The authors may consider removing the figure and might rather provide a more extensive Table 1.

Response: Figure 2 and table 1 are different. Figure 2 depicts the air-trapping feature of acute asthma, which causes detrimental effects during ventilation. In comparison, table 1 gives the clinical features that indicate a patient may require immediate intubation. 

#30 For all strategies the authors are discussing, a systematic display of studies (e.g. a large table) might be helpful. I would appreciate if the authors could provide such a table that provides key facts in a more systematic fashion. Especially when the evidence is limited it might be helpful to see for the readers that there are just 1-2 studies showing that x is not very much backed up by literature in a table.

Response: Thank you. While we agree this would be suitable for a systematic review, this is a narrative review aimed to provide available evidence than all studies conducted for a specific intervention. This is beyond the scope of this review. 

Conclusions

#31 Before the conclusion section, I would appreciate if the authors could state limitations of their review. Limitations may include literature access (I assume MEDLINE was used for the search?), the non-systematic review, as well as difficulty to compare studies because of large heterogeneity across the study groups.

Response: We introduced a paragraph to address this comment (line 861-866).

#32 As the authors clearly suggest future studies, I want to challenge the authors on the following: It would be very helpful (but not mandatory) to provide a design for a randomized controlled trial. The authors may select a treatment option that they think is most urgent to be better understood (and perhaps most promising) and should provide some recommendations on the setting, the population as well as the definitions of exposure and endpoints.

Response: Thank you for this comment. While this is an interesting request, it is beyond the scope of this review.

Again, I thank the authors for this interesting and well-written review and look forward to receiving a revised version. Thank you very much. 

Round 2

Reviewer 2 Report

Comments and Suggestions for Authors

The authors made substantial revisions to the text and figures, and have sufficiently addressed my comments. Thank you. I believe this version is now acceptable for publication in Applied Sciences.

Reviewer 3 Report

Comments and Suggestions for Authors

Thank you for providing a revised version of this review. I have no further comments.